# MP-GFormer: A 3D-Geometry-Aware Dynamic Graph Transformer Approach for Machining Process Planning

## Abstract

Machining process planning (MP) is inherently complex due to structural and geometrical dependencies among part features and machining operations. A key challenge lies in capturing dynamic interdependencies that evolve with distinct part geometries as operations are performed. Machine learning has been applied to address challenges in MP, such as operation selection and machining sequence prediction. Dynamic graph learning (DGL) has been widely used to model dynamic systems, thanks to its ability to integrate spatio-temporal relationships. However, in MP, while existing DGL approaches can capture these dependencies, they fail to incorporate three-dimensional (3D) geometric information of parts and thus lack domain awareness in predicting machining operation sequences. To address this limitation, we propose MP-GFormer, a 3D-geometry-aware dynamic graph transformer that integrates evolving 3D geometric representations into DGL through an attention mechanism to predict machining operation sequences. Our approach leverages StereoLithography surface meshes representing the 3D geometry of a part after each machining operation, with the boundary representation method used for the initial 3D designs. We evaluate MP-GFormer on a synthesized dataset and demonstrate that the method achieves improvements of 24% and 36% in accuracy for main and sub-operation predictions, respectively, compared to state-of-the-art approaches.

## 1 Introduction

Machining process planning (MP) is a systematic activity that converts raw materials into a finished part through a sequence of machining operations (e.g., milling, turning, and drilling) Afif & Sarhan (2025). MP determines the order of operations, defining tool paths, choosing cutting tools, and setting parameters Afif & Sarhan (2025). It serves as a bridge between computer-aided design (CAD) and computer-aided manufacturing (CAM), enabling CAD-based complex product geometries to be translated into CAM-based machining steps Wang et al. (2024), ensuring production efficiency, productivity, and final part quality Marzia et al. (2023). Traditional MP heavily relies on expert knowledge, which constrains efficient decision-making due to limited automation Besharati-Foumani et al. (2019). Machine learning (ML) techniques have now emerged as enablers of MP automation, learning process patterns from data and reducing the need for manual decision-making. In particular, recent ML studies in MP have leveraged advanced methods such as graph neural networks (GNNs) and large language models to model complex machining processes and generate process knowledge Wang et al. (2024); Xu et al. (2025), making a shift toward automated planning.

Dynamic graph learning (DGL) has been widely used for modeling dynamic systems thanks to its ability to integrate spatio-temporal relationships in various applications, including social network prediction, recommender systems, and traffic forecasting Yang et al. (2024). Dynamic graphs represent entities as nodes and represent their interactions as edges over time Yu et al. (2023). Motivated by their ability to learn long-range and complex dependencies, transformers Vaswani et al. (2017) have increasingly been adopted in DGL, capturing evolving structural and temporal relationships Peng et al. (2025). Recent studies have begun to explore the application of graph transformer architectures in manufacturing, showing the capabilities of attention mechanisms in capturing part-geometric relationships Dai et al. (2025) and learning process sequences Ma et al. (2022).

MP remains highly complex due to the evolving dependencies between part geometries and successive machining operation stages, where each operation in sequence introduces new dynamic interdependencies Xia et al. (2018); Zhang (1994). In MP, 3D geometric information of parts serves as the essential domain knowledge, as the shape and features of a part directly influence the selection, sequencing, and feasibility of machining operations. Dynamic graph modeling shows high potential for capturing the evolving dependencies between part geometries and machining operations within machining processes, while transformer architectures are well-suited to learn the sequential relationships—together enabling a spatio-temporal representation of MP that integrates part geometries and sequential machining operations. However, transformer-based DGL approaches in the area remain limited. Current methods are not domain-aware, as they do not explicitly incorporate 3D geometric domain information of parts into their frameworks Yu et al. (2023). This lack of domain-aware graph Transformer methods limits their ability to capture the manufacturing-specific constraints that arise from part geometry.

These challenges necessitate the development of a transformer-based DGL approach that can: (1) incorporate geometric information to guide process planning and decision-making and (2) capture evolving dependencies between part 3D geometries and sequential machining operations. Motivated by these, in this work, we propose a transformer-based DGL framework for machining operations prediction and sequence planning, termed MP-GFormer, for capturing 3D geometric and spatio-temporal dependecies of machined parts. The proposed method utilized a graph attention network (GAT) to explicitly model the dependencies between nodes and edges, enhancing relational reasoning across parts' 3D geometric entities. Second, an attention mechanism captures geometry-aware dependecies allowing the model to leverage geometric knowledge. Finally, a transformer decoder is employed to capture dependencies between the part's geometry and machining operations for predicting operation sequences.

## 2 RELATED WORKS

In this section, we review state-of-the-art research in transformer-based DGL and then discuss its emerging applications in MP.

**Transformer-Based Dynamic Graph Learning:** Recently, transformer-based dynamic graph representation learning has been widely studied in various fields. Sankar et al. (2018) proposed DySAT, a dynamic self-attention network to learn node representation by capturing structural and temporal properties. Xu et al. (2020) proposed TGAT, a temporal graph attention framework to learn temporal-topological features in a graph using a self-attention mechanism. Yu et al. (2023) proposed DyGFormer, a unified Transformer architecture designed for sequential dynamic graph learning, in which neighborhood co-occurrence features are incorporated to effectively model evolving interactions. Karmim et al. (2024) proposed SLATE, a fully connected transformer architecture to capture spatio-temporal dependencies utilizing a supra-Laplacian encoding approach. Wang et al. (2025) proposed CorDGT, a dynamic graph transformer with correlated spatio-temporal positional encoding for node representation learning. Peng et al. Peng et al. (2025) proposed TIDFormer, a dynamic graph transformer architecture that captures temporal and interactive dynamics.

**Graph Learning and Transformers in Machining Process Planning:** In MP, ML has been widely used to address different challenges, including operation selection, toolpath optimization, machining sequence prediction, and control. In particular, recent studies have increasingly explored graph modeling to address the complexity of MP to capture and preserve topological and geometrical relationships. For instance, Zhang et al. (2025) developed a graph encoder-decoder framework to predict the sequence of operations. Wang et al. (2024) proposed a graph convolutional neural network for feature machining operations prediction. Hussong et al. (2025) proposed MaProNet, a GAT-based architecture to capture topological and geometrical information and to predict manufacturing processes. Thanks to their long-range sequential dependency learning through the attention mechanism, transformers are applied in MP to predict machining process sequences. Dai et al. (2025) proposed BRepFormer, a transformer-based architecture to capture complex geometrical features from BRep for machining feature recognition. Maqueda et al. (2025) proposed DeepMS, a transformer-based method for automated prediction of machining operations and their sequences utilizing final part geometries.

However, the current DGL still remains limited in MP, for the current approaches do not explicitly incorporate 3D geometry-rich domain information of parts into their framework, and they primarily focus on learning sequential graph structures Yu et al. (2023). The lack of domain-aware graph transformer approaches limits their ability to capture the complexity of MP that arises from a part's geometry. A machined part typically contains multiple interacting features, such as slots, holes, and pockets Verma & Rajotia (2010), where a sequence of operations is performed, thereby transforming the part's geometry. The transformed geometry influences the next allowable operations and determines the possibility of the next operation in sequence. A key challenge lies in capturing those dynamic interdependencies that evolve with distinct part geometries as operations are performed. Additionally, current DGL and transformer approaches in MP have not fully utilized the capabilities of both methods. To address these challenges, we frame MP as a transformer-based DGL problem to capture spatio-temporal and geometrical dependencies of parts, which will be elaborated in Section three.

## 3 METHODOLOGY

### 3.1 PROBLEM FORMULATION

To address the limitations, we propose MP-GFormer, a 3D-geometry-aware dynamic graph transformer that integrates evolving 3D geometric representations into DGL through an attention mechanism to predict machining operation sequences, as shown in Figure 1.

We define each transition of the geometry in the machining process after an operation is performed as a *Process Graph* in Equation 1:

$$G^t = (V^t, E^t, H^t) \tag{1}$$

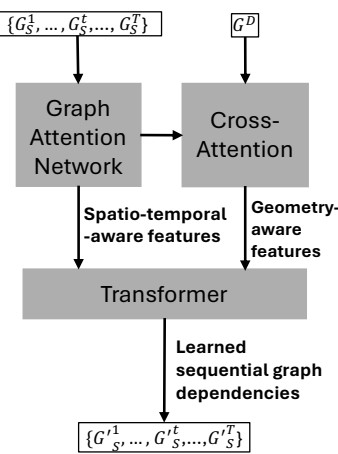

where $V^t$ and $E^t$ denote the nodes and edges at timestep $t$, and $H^t \in \mathbb{R}^{N_t \times d}$ is the graph feature matrix, indicating geometry and spatio-temporal attributes of $V^t$ and $E^t$, where $N$ and $d$ are the number of nodes and features, respectively. The machining process is thus represented as a sequence of *Process Graphs* as written in Equation 2:

$$\mathcal{S} = \{G^1, G^2, \dots, G^T\} \tag{2}$$

where $T$ is the total number of machining operations performed on a machined part. We also define the initial 3D geometry design of the part as a static *Design Graph* as shown in Equation 3:

$$G^D = (V^D, E^D, H^D) \tag{3}$$

where $V^D$ and $E^D$ denote the nodes and edges of the part's design, and $H^D \in \mathbb{R}^{N_D \times c}$ is the graph feature matrix, indicating design geometry attributes of $V^D$ and $E^D$, where $N$ and $c$ are the number of nodes and features, respectively.

We employ a transformer architecture to capture geometry and spatio-temporal dependencies between graphs in sequence to learn the function $f$ as written in Equation 4:

Figure 1: The proposed framework integrates a GAT, an attention mechanism, and a transformer to capture and learn the spatio-temporal and geometry-aware dependencies in MP.

$$f : (\mathcal{S}, G^D) \mapsto \mathcal{S}^L = \{G^{L_1}, G^{L_1}, \dots, G^{L_T}\} \tag{4}$$

where $G^{L_t} = (V^{L_t}, E^{L_t}, H^{L_t})$ contains learned geometry and spatio-temporal dependencies between graphs in sequence in a machining process. The learned dependencies are later used for predicting machining operation sequences through the proposed MP-GFormer. Figure 2 indicates the details of the architecture. MP-GFormer contains three main stages: a) Encoder, b) Transformer, and c) Classifier.

### 3.2 ENCODER ARCHITECTURE

The encoder is designed to capture geometry and spatio-temporal features from both *Process Graphs* and the *Design graph* It consists of graph attention encoding, temporal encoding, cross-attention fu-

Figure 2: The MP-GFormer architecture. Sequential machining process graphs and the initial design graph are used as inputs. The architecture consists of three main stages: (a) an encoder, where node and edge are mapped into a latent space using a GAT encoder, and a cross-attention mechanism captures interdependencies between process and design graphs; (b) a Transformer decoder that learns dependencies across sequential graphs, machining operations, and their interactions through three attention mechanisms; and (c) a classifier that predicts machining operations.

sion, feature embedding, attention pooling, positional encoding, and label embedding. First, we apply a GAT Veličković et al. (2017) to capture and learn node and edge features of each process graph $G^t$ and the design graph $G^D$ at timestep $t$. Node $H^t = \{\mathbf{h}_1^t, \ldots, \mathbf{h}_{N_t}^t\}$ and $H^D = \{\mathbf{h}_1^D, \ldots, \mathbf{h}_{N_D}^D\}$ and edge $R^t = \{\mathbf{r}_{ij}^t \mid (i,j) \in \mathcal{E}^t\}$ and $R^D = \{\mathbf{r}_{kl}^D \mid (k,l) \in \mathcal{E}^D\}$ embeddings are projected into a latent space as written in Equation 5 for *Process Graphs* and *Design Graph*:

$$\mathbf{z}_i^t = W_n \mathbf{h}_i^t, \quad \mathbf{z}_j^D = W_n \mathbf{h}_j^D, \quad \mathbf{e}_{ij}^t = W_e \mathbf{r}_{ij}^t, \quad \mathbf{e}_{kl}^D = W_e \mathbf{r}_{kl}^D, \quad W \in \mathbb{R}^{d' \times d} \tag{5}$$

Later, attention scores are computed between neighboring nodes and their edges in both graphs as seen in Equation 6 and normalized across neighbors as written in Equation 7:

$$s_{ij}^t = a\left(\mathbf{z}_i^t, \mathbf{z}_j^t, \mathbf{e}_{ij}^t\right), \quad s_{kl}^D = a\left(\mathbf{z}_k^D, \mathbf{z}_l^D, \mathbf{e}_{kl}^D\right) \tag{6}$$

$$\alpha_{ij}^t = \frac{\exp(s_{ij}^t)}{\sum_{k \in \mathcal{N}(i)} \exp(s_{ik}^t)}, \quad \alpha_{kl}^D = \frac{\exp(s_{kl}^D)}{\sum_{m \in \mathcal{N}(k)} \exp(s_{km}^D)} \tag{7}$$

We updated the embeddings via attention-weighted aggregation utilizing multi-head attention as written in Equations 8, 9, and 10:

$$\mathbf{h}_i^{t'} = \sigma\left(\sum_{j \in \mathcal{N}(i)} \alpha_{ij}^t \mathbf{z}_j^t\right), \quad \mathbf{h}_k^{D'} = \sigma\left(\sum_{l \in \mathcal{N}(k)} \alpha_{kl}^D \mathbf{z}_l^D\right) \tag{8}$$

$$\mathbf{h}_i^{t'} = \sigma\left(\frac{1}{M}\sum m = 1^M \sum_{j \in \mathcal{N}(i)} \alpha^{t,(m)}ij\left(W_n^{(m)}\mathbf{h}_j^t + W_e^{(m)}\mathbf{r}^t ij\right)\right) \tag{9}$$

$$\mathbf{h}_k^{D'} = \sigma\left(\frac{1}{M}\sum m = 1^M \sum_{l \in \mathcal{N}(k)} \alpha^{D,(m)}kl\left(W_n^{(m)}\mathbf{h}_l^D + W_e^{(m)}\mathbf{r}^D kl\right)\right) \tag{10}$$

where $M$ is the number of heads. The outputs are the learned node and edge features $H^{t'}$ for each *Process Graph* $G^t$ and $H^{D'}$ for the *Design Graph* $G^D$. As each sequence $S$ may contain a different number of timesteps, we employ a linear encoder to project the time features into the same latent space dimension as the graph features, as written in Equation 11:

$$\mathbf{h}_{\text{time}}^t = W_t \cdot \text{onehot}(t), \quad W_t \in \mathbb{R}^{d' \times T} \tag{11}$$

To capture the interdependency between the geometry of the *Process Graphs* $G^t$ and the initial design geometry $G^D$, we employ a cross-attention mechanism Vaswani et al. (2017), where $G^t$ acts as query and $G^D$ serves as key and value as shown in Equations 12 and 13:

$$Q = H^{t'} W_Q, \quad K = H^{D'} W_K, \quad V = H^{D'} W_V \tag{12}$$

$$\tilde{H}^t = \text{softmax}\left(\frac{QK^\top}{\sqrt{d_k}}\right) V \tag{13}$$

where $\tilde{H}^t$ are geometry-aware fused feature embeddings.

Later, we combined all encoded features into a unified representation, as written in Equation 14:

$$F^t = \phi\left(\tilde{H}^t, H^{t'}, \mathbf{h}^t_{\text{time}}\right) \tag{14}$$

where $\phi(\cdot)$ is a feedforward projection. Later, we aggregate node-level embeddings into a graph-level representation of each process graph using attention pooling Li et al. (2015), as defined in Equation 15:

$$G'^t = \sum_{i=1}^{N_t} \beta_i^t F_i^t, \tag{15}$$

where $\beta_i^t$ is the attention weight assigned to node $i$ at timestep $t$. The output of this step is a sequence of graph embeddings for each sequence $S$, as written in Equation 16:

$$\mathcal{S}' = \{G'^1, G'^2, \ldots, G'^T\} \tag{16}$$

### 3.3 TRANSFORMER ARCHITECTURE

The Transformer learns sequential graph embeddings to capture the geometric and spatio-temporal dependencies across different machining processes. To this end, we employ a transformer decoder architecture with three attention mechanisms to capture: (i) dependencies between *Process Graphs* in the sequence, (ii) dependencies between operations in the sequence, and (iii) cross-dependencies between graphs and operations. We applied positional encoding to preserve the order of the graphs in sequence (Equation 17).

$$\hat{G}^t = G'^t + \text{PE}(t) \tag{17}$$

Additionally, each machining operation, as the label of interest, is mapped to a learnable vector embedding (Equation 18).

$$L_{y_t} = \text{Em}[y_t] \in \mathbb{R}^{d'} \tag{18}$$

where $L_{y_t}$ denotes the learned operation embedding at timestep $t$.

We also define the set of past operation embeddings up to step $t-1$ as in Equation 19:

$$L_Y = [L_{y_1}, L_{y_2}, \ldots, L_{y_{t-1}}]^\top \in \mathbb{R}^{(t-1) \times d'} \tag{19}$$

We first apply a masked self-attention mechanism over $L_Y$, as defined in Equation 20, 21, and 22 :

$$Q_{L_Y} = L_Y W_Q^{(L_Y)}, \quad K_{L_Y} = L_Y W_K^{(L_Y)}, \quad V_{L_Y} = L_Y W_V^{(L_Y)} \tag{20}$$

$$A_{L_Y} = \text{Softmax}\left(\frac{Q_{L_Y} K_{L_Y}^\top}{\sqrt{d}} + CM\right) \tag{21}$$

$$L'_Y = A_{L_Y} V_{L_Y} \tag{22}$$

where $CM$ is a causal mask.

Second, a temporal self-attention mechanism is applied to the sequence of *Process Graph* embeddings, defined as $\hat{S}_{1:t} = [\hat{G}^1, \hat{G}^2, \ldots, \hat{G}^t]$, as written in Equations 23, 24, and 25: Self-attention is then computed over the sequence $\hat{S}_t$, with all projections defined as

$$Q_{\hat{S}_{1:t}} = \hat{S}_{1:t} W_Q^{(\hat{S}_{1:t})}, \quad K_{\hat{S}_{1:t}} = \hat{S}_{1:t} W_K^{(\hat{S}_{1:t})}, \quad V_{\hat{S}_{1:t}} = \hat{S}_{1:t} W_V^{(\hat{S}_{1:t})} \tag{23}$$

$$A_{\hat{S}_{1:t}} = \text{Softmax}\left(\frac{Q_{\hat{S}_t} K_{\hat{S}_{1:t}}^\top}{\sqrt{d}}\right) \tag{24}$$

$$\hat{S}'_{1:t} = A_{\hat{S}_{1:t}} V_{\hat{S}_{1:t}} \tag{25}$$

Finally, the label representation $L'^t_Y$ attends to the graph sequence $\hat{S}'_{1:t}$ via cross-attention, as written in Equations 26, 27, and 28:

$$Q_t = W_Q^{(c)} L'^t_Y, \quad K_c = \hat{S}'_t W_K^{(c)}, \quad V_c = \hat{S}'_t W_V^{(c)} \tag{26}$$

$$A_{cross} = \text{Softmax}\left(\frac{Q_t K_c^\top}{\sqrt{d}}\right) \tag{27}$$

$$G^{L_t} = A_{cross} V_c \tag{28}$$

where $G^{L_t}$ denotes the learned graph embeddings at timestep $t$, which contains the geometry and spatio-temporal dependencies between process and design graphs for a machined part.

## 3.4 Classifier Architecture

At each timestep $t$, the classifier predicts the machining operation from the learned graph embedding $G^{L_t}$. The logits are computed as in Equation 29:

$$y_t = W_o G^{L_t} + b_o \tag{29}$$

where $W_o$ and $b_o$ are weight and bias parameters for the operation classifier. Later, the machining operation is predicted as shown in Equation 30:

$$\hat{y}_t = \arg\max\left(\text{Softmax}(y_t)\right) \tag{30}$$

The model is trained using cross-entropy loss as written in Equation 31:

$$\mathcal{L}_t = -\log p(y_t \mid G^{L_t}) \tag{31}$$

The overall training objective is the average cross-entropy across all timesteps as shown in Equation 32:

$$\mathcal{L} = \frac{1}{T} \sum_{t=1}^{T} \mathcal{L}_t \tag{32}$$

## 4 Data Set

The data contains three types of files in this work: (1) the Stereolithography files (STL), representing geometry as a faceted triangle mesh, (2) the Boundary representation (BRep) files, which provide a comprehensive description of the topology and geometry of the part design, and (3) the textual files indicating the operations and process information.

### 4.1 Data Collection

Synthetic data was generated based on real-world example images for machining, which can be organized into two classes: (1) simple and (2) complex geomtry, shown in Figure 3. Each geometry was parameterized for a) the length, width, and height of the part design and b) the pocket/hole location and sizes. By running a script through the range of values of each parameter and updating the process plan, we generated 2991 valid data sets. Figure 8 indicates samples of generated synthesized data. Please refer to Appendix A.1 for more details.

### 4.2 Graph Generation

We constructed two types of graphs: (1) sequential geometry graphs derived from STL files corresponding to each manufacturing main and sub-operation, and (2) initial design graphs derived from BRep models of the manufactured part.

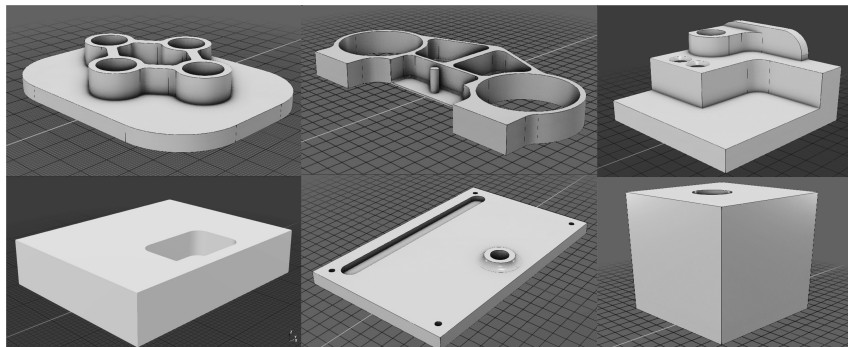

Figure 3: Samples of parts geometries used for synthesized data generation: (1) simple geometry (bottom row) and (2) complex geometry (top row).

**STL-based Graphs.** Each STL file represents the geometry of a machined part at a specific timestep. We treat each triangle face as a node $v_i \in V^t$, with node features $\mathbf{h}_i^{\text{STL}} = [\text{centroid}(v_i), \text{ normal}(v_i), \text{ area}(v_i), \text{ perimeter}(v_i), \text{ edge length}(v_i), \text{ angles}(v_i), \text{ angle types}(v_i), \text{ compactness}(v_i)] \in \mathbb{R}^8$. Edges $e_{ij} \in E^t$ are created between two triangle faces that share a common edge. Each edge vector contains geometric distances and curvature features as $\mathbf{x}_{ij}^{\text{STL}} = [\text{centroid distance}(v_i, v_j), \text{ normal distance}(v_i, v_j), \text{ shared-edge length}(e_{ij}), \text{ edge midpoint}(e_{ij}), \text{ dihedral angle}(e_{ij})] \in \mathbb{R}^5$.

**BRep-based Graphs.** We constructed graphs directly from BRep models, indicating the design geometry of machined parts. Each face of the B-Rep model is treated as a node, with features defined as $\mathbf{h}_i^{\text{BRep}} = [\text{surface type}(f_i), \text{ area}(f_i), \text{ UV-parameters}(f_i), \text{ 3D centroid}(f_i)] \in \mathbb{R}^{d_{\text{BRep}}}$. Each edge corresponds to adjacencies between faces in the B-Rep model. For each edge, we extract shape-aware features as $\mathbf{x}_{ij}^{\text{BRep}} = [\text{shared-edge length}(e_{ij}), \text{ dihedral angle}(e_{ij}), \text{ edge curve type}(e_{ij})] \in \mathbb{R}^3$.

The final graph data, therefore, consists of a sequence of STL graphs $\mathcal{S} = \{G_{\text{STL}}^1, G_{\text{STL}}^2, \ldots, G_{\text{STL}}^T\}$ and a design-based BRep graph $G^{\text{BRep}}$ that together provide geometry-based spatio-temporal information of a sequence of machining operations. Figure 9 indicates sample data inputs for the graph generation step. Later, the generated graphs are used as the input for the proposed methodology, MP-GFormer, for the main and sub-operation sequence classification task. Figure 4 indicates the machining operation prediction utilizing STL and BRep graphs as the inputs for MP-GFormer, which captures the dependency between sequential STL graphs as well as the relationship between BRep and each STL graph to predict a sequence of operations.

## 5 EXPERIMENTS

**Hyperparameter Selection:** The data set contains 2991 graph sequences with main and sub-operations as labels of interest. Figure 10 indicates the number of unique labels in the dataset. We utilized 80% and 20% for training and testing, respectively. Tables 1 and 2 indicate the results of training and testing for different hyperparameter values. Figure 5 shows the training and testing loss curves for main and sub-operation predictions using the best hyperparameter settings. As a more detailed qualitative analysis, we calculated t-SNE values for the learned graph embeddings during training over different epochs with the best hyperparameter values, as shown in Figure 6. Each data point corresponds to one graph, and each different color corresponds to an operation class. The results indicate that graphs within the same operation classes are well-separated and clustered for both operations, reflecting the transformer's capability to capture the structural and sequential dependencies of graphs in MP.

**Ablation study:** We considered two different variants of the method for the ablation study. Table 3 indicates the ablation study results. First, we replaced the GAT encoder with a neural network (NN) encoder for nodes and edges feature encoding of the first layer of the architecture. The results

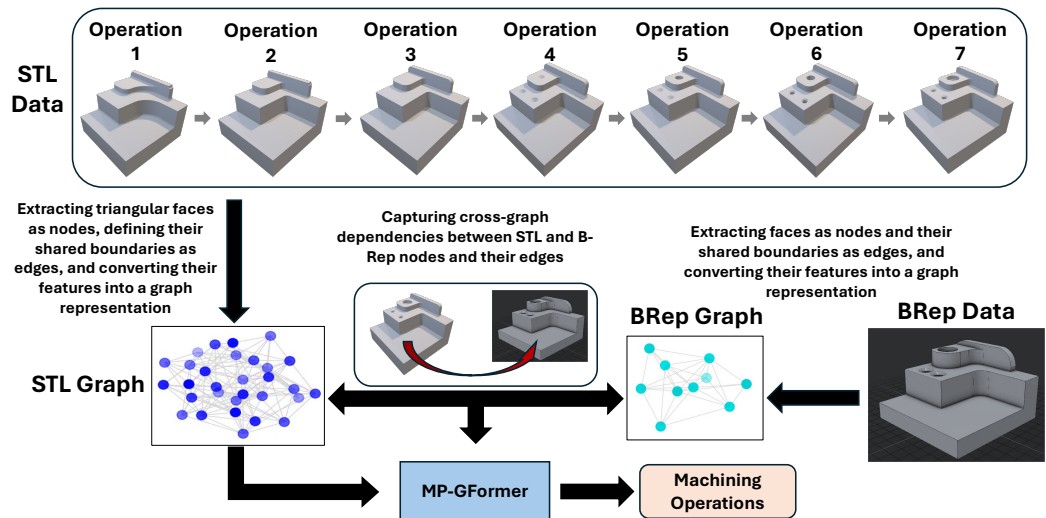

Figure 4: Machining operation prediction via MP-GFormer. The case study utilizes STL geometry data and the BRep model as inputs, which are converted into graph representations. These graph-based inputs are then processed by MP-GFormer to predict sequential machining operations.

Table 1: Results under Different Hyperparameter Settings for Training (Accuracy, F1, Precision, Recall)

| Hyperparameters | Main Operation | | | | Sub Operation | | | | Joint Operations | | | |
|---|---|---|---|---|---|---|---|---|---|---|---|---|
| | Acc | F1 | Prec | Rec | Acc | F1 | Prec | Rec | Acc | F1 | Prec | Rec |
| Batch=8, LR=0.01, Epochs=5 | 0.69 | 0.64 | 0.63 | 0.69 | 0.74 | 0.69 | 0.71 | 0.74 | 0.62 | 0.55 | 0.48 | 0.62 |
| Batch=16, LR=0.01, Epochs=5 | 0.74 | 0.73 | 0.77 | 0.74 | 0.74 | 0.69 | 0.71 | 0.74 | 0.68 | 0.62 | 0.59 | 0.68 |
| Batch=128, LR=0.01, Epochs=5 | 0.71 | 0.70 | 0.72 | 0.71 | 0.74 | 0.69 | 0.71 | 0.74 | 0.64 | 0.59 | 0.57 | 0.64 |
| Batch=16, LR=0.001, Epochs=5 | 0.72 | 0.72 | 0.75 | 0.72 | 0.74 | 0.70 | 0.71 | 0.74 | 0.65 | 0.60 | 0.59 | 0.65 |
| **Batch=128, LR=0.001, Epochs=20** | **0.74** | **0.72** | **0.78** | **0.74** | **0.70** | **0.66** | **0.68** | **0.70** | **0.63** | **0.60** | **0.60** | **0.63** |
| Batch=256, LR=0.001, Epochs=20 | 0.71 | 0.70 | 0.73 | 0.71 | 0.67 | 0.63 | 0.66 | 0.67 | 0.59 | 0.55 | 0.55 | 0.59 |

Table 2: Results under Different Hyperparameter Settings for Test (Accuracy, F1, Precision, Recall)

| Hyperparameters | Main Operation | | | | Sub Operation | | | | Joint Operations | | | |
|---|---|---|---|---|---|---|---|---|---|---|---|---|
| | Acc | F1 | Prec | Rec | Acc | F1 | Prec | Rec | Acc | F1 | Prec | Rec |
| Batch=8, LR=0.01, Epochs=5 | 0.77 | 0.77 | 0.79 | 0.77 | 0.33 | 0.18 | 0.12 | 0.33 | 0.33 | 0.20 | 0.15 | 0.33 |
| Batch=16, LR=0.01, Epochs=5 | 0.77 | 0.77 | 0.79 | 0.77 | 0.27 | 0.15 | 0.10 | 0.27 | 0.27 | 0.16 | 0.11 | 0.27 |
| Batch=128, LR=0.01, Epochs=5 | 0.76 | 0.76 | 0.78 | 0.76 | 0.25 | 0.13 | 0.09 | 0.25 | 0.25 | 0.15 | 0.10 | 0.25 |
| Batch=16, LR=0.001, Epochs=5 | 0.76 | 0.76 | 0.78 | 0.76 | 0.25 | 0.13 | 0.09 | 0.25 | 0.25 | 0.15 | 0.10 | 0.25 |
| **Batch=128, LR=0.001, Epochs=20** | **0.78** | **0.75** | **0.77** | **0.75** | **0.70** | **0.63** | **0.60** | **0.61** | **0.61** | **0.60** | **0.61** | **0.62** |
| Batch=256, LR=0.001, Epochs=20 | 0.72 | 0.73 | 0.75 | 0.72 | 0.60 | 0.63 | 0.60 | 0.61 | 0.59 | 0.34 | 0.35 | 0.25 |

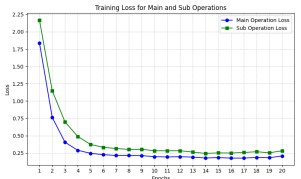

(a) Training loss over 20 epochs for Main and Sub Operations

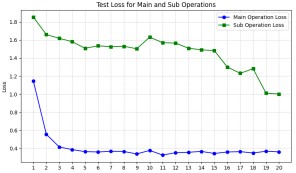

(b) Test loss over 20 epochs for Main and Sub Operations

Figure 5: Loss curves.

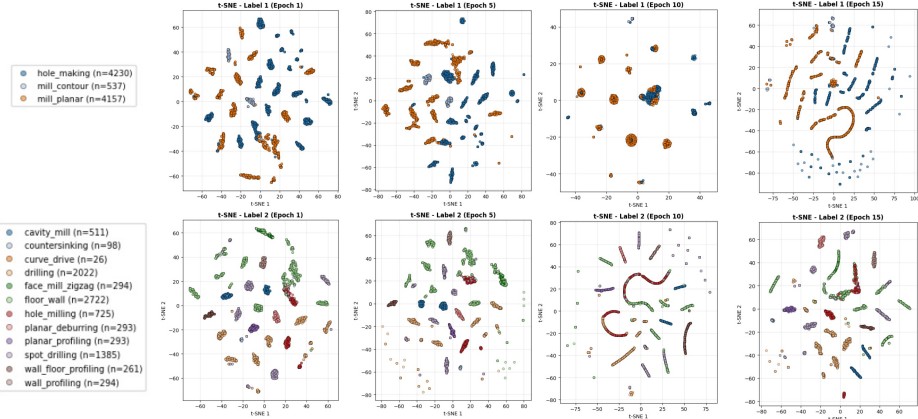

Figure 6: t-SNE visualization of learned graph embeddings across training epochs. Each color corresponds to an operation class, and each point represents one graph sample. As training progresses, samples of each class become more separable and form clusters.

indicated a decline in accuracy for both training and testing. The NN encoder ignores the interactions between a node and its neighboring nodes, thereby losing important structural information from the graph. Secondly, we employed a transformer encoder for sequential graph learning instead of a decoder layer, which resulted in a significant improvement in performance. This can be explained by the encoder's parallel graph processing ability, thereby having access to the future graph structures and labels when predicting the current label. This may cause the model to benefit from data leakage during both training and testing, as it leverages information from future steps that should not be available for predicting the current label.

Table 3: Ablation Study Results (Train and Test: Accuracy, F1, Precision, Recall)

| Method | Main Operation | | | | | | | | Sub Operation | | | | | | | | Joint Operations | | | | | | | |
|---|---|---|---|---|---|---|---|---|---|---|---|---|---|---|---|---|---|---|---|---|---|---|---|---|
| | Train | | | | Test | | | | Train | | | | Test | | | | Train | | | | Test | | | |
| | Acc | F1 | Prec | Rec | Acc | F1 | Prec | Rec | Acc | F1 | Prec | Rec | Acc | F1 | Prec | Rec | Acc | F1 | Prec | Rec | Acc | F1 | Prec | Rec |
| NN-Based | 0.64 | 0.51 | 0.54 | 0.50 | 0.66 | 0.49 | 0.53 | 0.50 | 0.47 | 0.20 | 0.22 | 0.21 | 0.50 | 0.21 | 0.23 | 0.24 | 0.42 | 0.11 | 0.13 | 0.11 | 0.44 | 0.15 | 0.16 | 0.15 |
| Encoder-based | 0.94 | 0.93 | 0.94 | 0.93 | 0.98 | 0.97 | 0.98 | 0.97 | 0.90 | 0.86 | 0.87 | 0.85 | 0.97 | 0.93 | 0.93 | 0.94 | 0.88 | 0.58 | 0.60 | 0.57 | 0.96 | 0.86 | 0.86 | 0.86 |
| MP-GFormer | 0.78 | 0.75 | 0.77 | 0.75 | 0.78 | 0.75 | 0.77 | 0.75 | 0.70 | 0.66 | 0.68 | 0.70 | 0.70 | 0.63 | 0.60 | 0.61 | 0.63 | 0.60 | 0.60 | 0.63 | 0.61 | 0.60 | 0.61 | 0.0.62 |

**Benchmark Study:** We compared the proposed method with two DGL approaches from the literature. First, with DyGFormer Yu et al. (2023), as the study was motivated by the method. Second, we compared the method with DiffPool Ying et al. (2018), a hierarchical graph representation learning approach. We trained both models with the same dataset used for the case study. Table 4 indicates the comparative study results. MP-GFormer achieves improvements of 24% and 36% in accuracy for main and sub-operation predictions, respectively, as compared to DyGFormer, and 24% and 20% as compared to DiffPool.

Table 4: Benchmark Study Results (Accuracy, F1, Precision, Recall) for Training and Testing

| | DyGFormer | | | | | | | | DiffPool | | | | | | | | MP-GFormer | | | | | | | |
|---|---|---|---|---|---|---|---|---|---|---|---|---|---|---|---|---|---|---|---|---|---|---|---|---|
| | Train | | | | Test | | | | Train | | | | Test | | | | Train | | | | Test | | | |
| | Acc | F1 | Prec | Rec | Acc | F1 | Prec | Rec | Acc | F1 | Prec | Rec | Acc | F1 | Prec | Rec | Acc | F1 | Prec | Rec | Acc | F1 | Prec | Rec |
| Main Operation | 0.52 | 0.36 | 0.35 | 0.37 | 0.54 | 0.36 | 0.40 | 0.40 | 0.52 | 0.39 | 0.34 | 0.52 | 0.54 | 0.39 | 0.31 | 0.54 | 0.78 | 0.75 | 0.77 | 0.75 | 0.78 | 0.75 | 0.77 | 0.75 |
| Sub Operation | 0.36 | 0.10 | 0.12 | 0.12 | 0.34 | 0.11 | 0.12 | 0.14 | 0.48 | 0.34 | 0.28 | 0.48 | 0.50 | 0.35 | 0.27 | 0.50 | 0.70 | 0.66 | 0.68 | 0.70 | 0.70 | 0.63 | 0.60 | 0.61 |

## 6 CONCLUSION

In this research, we introduced MP-GFormer, for predicting machining operation sequences, which utilized sequential machining process geometry graphs derived from STL, and the initial design geometry graph from the BRep model to capture spatio-temporal dependencies and geometry-aware relationships. The proposed method outperformed the baseline DGL approaches in predicting machining operation sequences. For future work, we aim to integrate diffusion models into DGL to enable the prediction of the 3D geometry of parts with operation sequences.

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

## A    APPENDIX

### A.1    DATA STRUCTURE

As has been stated, MP is inherently complex, with many different types of operations, tools, and parameter selections for a specific type of machining. In order to avoid exploding the solution space, we focus this study on planar CNC machining (2.5-axis machining). Within planar machining, we limit the operation type to the following 2-level hierarchy: Mill Planar (floor wall, wall floor profiling, wall profiling, face mill zigzag, planar profiling, planar deburring), Hole Making (drilling, spot drilling, hole milling, countersinking), and Mill Contour (cavity milling, curve drive). Furthermore, part design geometries are created in a Computer-Aided Design software and then processed via a Computer-Aided Machining software. All designs are assumed to be of the same material. Each data set consists of the following a) Input stock/blank geometry as stl, b) N-numbered In-process workpiece (IPW) as stl (N varies for different data set), c) N-numbered two tiered lables for each data set, d) Output Geometry as feature rich BRep.

### A.2    EXPERIMENTS

Figure 10 indicates the number of main and sub-operations in the synthesized dataset. There exist three main and twelve sub-operations as the labels of interest for the machining operation sequences classification task.

### A.3    LLM USAGE

LLMs were used for grammar checking, polishing of the manuscript text, and for identifying comparative methods in the benchmark study.

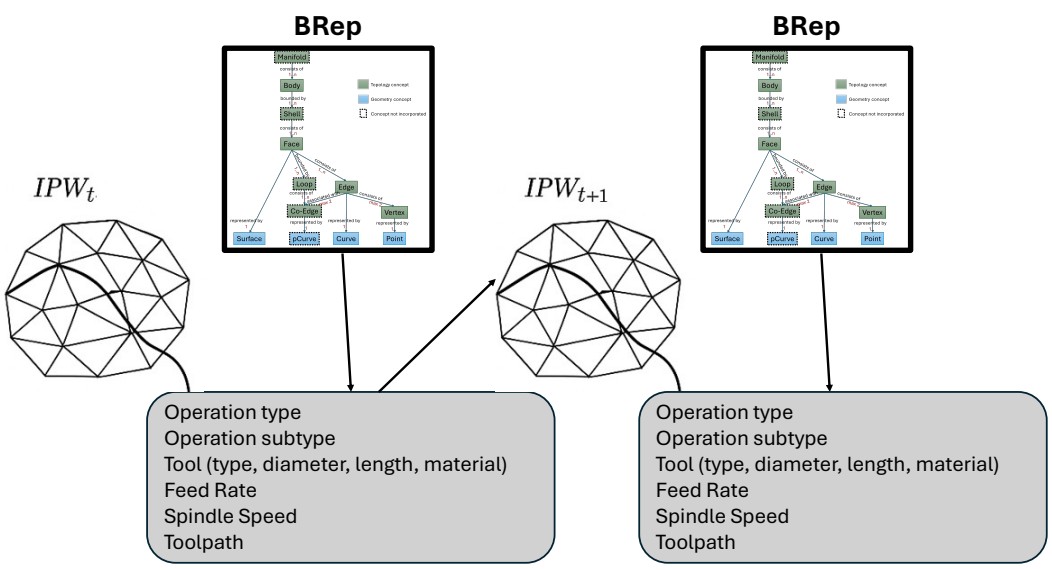

Figure 7: Raw Data Structure.

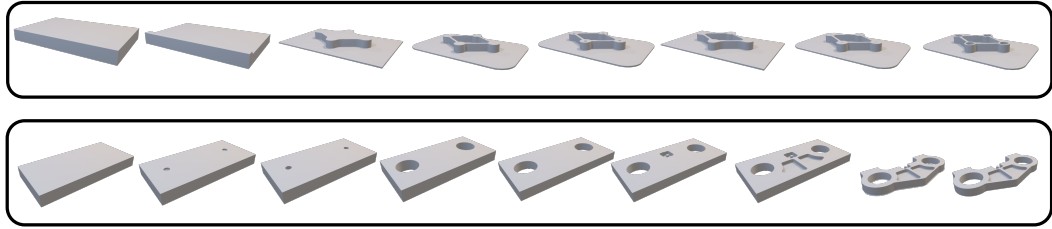

Figure 8: Samples of generated synthesized STL data for two different parts.

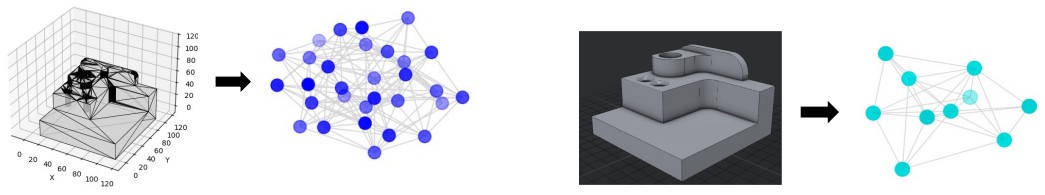

(a) STL Graph generation from triangles.      (b) BRep Graph generation from faces.

Figure 9: Graph Generation.

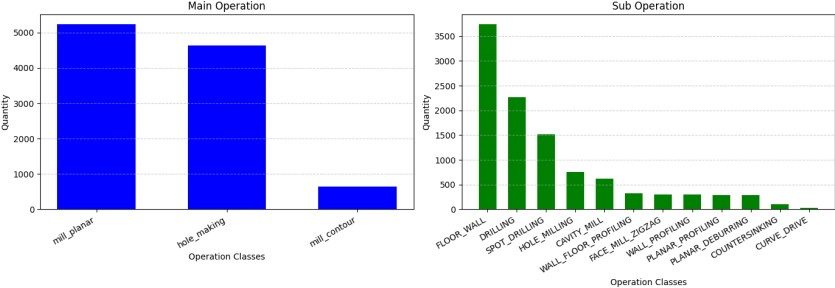

Figure 10: The labels distribution

