# OpenReview forum: "MP-GFormer: A 3D-Geometry-Aware Dynamic Graph Transformer Approach for Machining Process Planning"
_ICLR.cc/2026/Conference — ICLR 2026 Conference Withdrawn Submission_

### Official Review · Reviewer_ns4B · 2025-10-28

**Soundness:** 2
**Presentation:** 2
**Contribution:** 1
**Rating:** 2
**Confidence:** 4

**Summary:**

This paper introduces MP-GFormer, a dynamic graph transformer architecture designed for machining process planning (MP). The core idea is to leverage the evolving 3D geometry of a part as it undergoes a sequence of machining operations to predict the subsequent operation. The authors frame this process as a dynamic graph learning problem where each stage of the machining process is represented by a graph derived from its STL mesh. These Process Graphs are processed alongside a static Design Graph to capture both the temporal evolution and the original design intent. To this end, a combination of Graph Attention Networks, cross-attention and a transformer decoder are used. The model is evaluated on a synthetic dataset, where it outperforms existing baselines in the classification task of finding the next operation given a sequence of previous operations and meshes.

**Strengths:**

- The paper is generally well-written and easy to read. The motivation for the approach is clear and convincing.
- The paper introduces a novel and relevant formulation of machining process planning as a dynamic graph learning problem conditioned on some target design. This approach of modeling the sequence of manufacturing steps as an evolving series of graphs is an interesting and appropriate way to capture the problem's inherent temporal and geometric dependencies.
- The work addresses an important application in smart manufacturing. Automating machining process planning is a critical challenge, and successful solutions may have a large real-world impact across several industries.
- The method crucially proposes to explicitly utilize evolving 3D geometric information. By representing each machining step as a graph derived from STL/BRep files, the proposed model captures the physical constraints that govern the manufacturing process.

**Weaknesses:**

- The proposed architecture, MP-GFormer, is presented as a complex sequence of components (GAT encoders, cross-attention, transformer decoder, etc.) without sufficient justification for each design choice. The paper does not explain why this specific, convoluted combination of modules is necessary over simpler alternatives, such as a temporal GNN.
- The experimental evaluation does not convincingly support the paper's claims. The hyperparameter tuning and ablation studies do not sufficiently consider and justify the different design choices of the architecture. The experiments seem to be conducted on a single seed, which makes it difficult to assess the statistical significance of the resuls. The t-SNE visualization offers little novel insights, since the latent space of the method should be expected to cluster in what is essentially a classification task. Further, the evaluation seems to be constrained to predicting individual labels, rather than, e.g., performing a full sequence of operations on a novel part and qualitatively showing how it evolves over time.
- The method section is largely comprised of different equations relating to the architectures components. These equations offer little insight into *why* the architecture was chosen in the way it was, and what the novel and interesting aspects of it are. In its current presentation, it is difficult to assess where exactly the novelty of the method lies.
- While the application is and interesting novel, the architectural components themselves are standard building blocks from the GNN and Transformer literature. The contribution is more of an application of existing methods to a new domain rather than a fundamental advance in deep learning methodology. Additionally, the different parts of the architecture are not fully justified, and it is unclear how simpler approaches would perform.
- In terms of presentation, the paper’s figures somewhat overlap, and in particular Figure 1 and 2 seem like they could be combined. Figure 4 would be an excellent introductory figure that shows the task that is solved by the proposed method, but is only mentioned relatively late in the paper.

**Questions:**

1. Could the authors provide a clearer justification for the complexity of the MP-GFormer architecture? For example, is the cross-attention mechanism between the evolving process graph and the static design graph critical? What would be the performance of a much simpler model, perhaps a recurrent GNN, on this task?
2. Standard GATs do not typically handle edge features. Could the authors clarify which specific variant of GAT was used that incorporates edge features as described in Equation 5 and the subsequent equations?
3. Can the authors clarify the train/test split? Was the split performed at the level of entire machining sequences, or were some timesteps of a sequence used for training and others for testing? If the latter, does this cause data leakage problems, where the model is tested on intermediate steps it has previously been trained on?
4. Could the authors provide some qualitative examples of the model's predictions? For instance, an example of a full sequence prediction, including cases where the model succeeds and where it fails, would be very informative.

---

### Official Review · Reviewer_yDtm · 2025-10-31

**Soundness:** 2
**Presentation:** 3
**Contribution:** 2
**Rating:** 2
**Confidence:** 3

**Summary:**

This paper introduces MP-GFormer, a model that helps plan machining by predicting the next operation in the sequence (like “face mill,” “drill,” etc.). It reads two kinds of geometry: the initial CAD design (as a B-Rep graph) and the part’s shape after each step (as STL graphs). A graph encoder and a Transformer with cross-attention learn how the geometry changes over time and use that to choose the next operation label at each step. The authors train on a synthetic 2.5-axis (planar) dataset of 2,991 sequences and report clearly better accuracy than generic dynamic-graph baselines. The method is useful for high-level sequencing, but it does not output detailed machining instructions (no locations, tools, feeds/speeds, or toolpaths)—only the operation labels within a fixed class set.

**Strengths:**

The paper cleanly scopes machining process planning to macro-level sequencing and builds a geometry-aware model that fuses the initial B-Rep design with per-step STL geometry, using GAT encoders for local structure, cross-attention to inject design intent, and a causally masked Transformer decoder to respect step order. It defines concrete node/edge features (centroids, normals, dihedral angles, surface types, etc.), reports strong gains over dynamic-graph baselines on a standardized synthetic dataset, and includes an ablation that surfaces and mitigates potential look-ahead leakage—justifying the decoder choice. The embedding visualizations are interpretable (class clusters sharpen over training), and the data/graph construction pipeline is well documented, aiding reproducibility within the stated scope.

**Weaknesses:**

The method’s output is non-executable: it predicts only main/sub operation labels per step without binding them to concrete geometric targets or proposing tools, parameters, or toolpaths, so it cannot yield a CAM-ready plan. Empirical validation relies on a small, synthetic, parameterized dataset with no evidence of transfer to real shops, machines, fixtures, or tool libraries, which limits external validity. The graph granularity is misaligned with machining semantics—modelling each STL triangle as a node ties learning to meshing artifacts rather than stable manufacturing features (holes, pockets, walls), undermining feature grounding and downstream actionability. The ablation exposes a risk of future-step leakage (encoder seeing “the future”), highlighting that results are fragile unless causal masking is rigorously enforced and audited. Finally, the operation taxonomy is thinly specified and shop-specific variability or class imbalance is not addressed, raising concerns about portability and label robustness.

**Questions:**

Referring to the Weaknesses,
1. Can you output (operation label + target feature IDs/regions) at each step instead of labels only? If not, what blocks binding operations to concrete geometry?
2. Why model STL triangles as nodes instead of feature-level nodes (holes/pockets/walls)? How sensitive are results to STL tessellation (coarse vs. fine, different exporters) for the same part?
3. How do you guarantee no look-ahead during training/testing? Please report performance under graph-order permutation/shuffle and show a leakage unit test that would fail if future steps are accessible.
4. Provide the labelling policy, per-class metrics, and how you handled imbalance; how portable is this taxonomy across shops/CAM systems?
5. Do results hold on real CAM histories (machines, fixtures, tool libraries)? If not, can you run a small pilot? How would predictions change when conditioned on machine limits and a tool library?
6. How do you know this pre-defined class set is sufficient beyond your synthetic generator?

---

### Official Review · Reviewer_DpXJ · 2025-11-01

**Soundness:** 3
**Presentation:** 2
**Contribution:** 3
**Rating:** 6
**Confidence:** 2

**Summary:**

This paper proposes MP-GFormer, a 3D-geometry-aware dynamic graph transformer for predicting machining operation sequences in manufacturing process planning. The key innovation is explicitly incorporating 3D geometric domain knowledge by integrating STL surface meshes (representing part geometry after each operation) and BRep models (initial designs) into a dynamic graph learning framework through a combination of Graph Attention Networks, cross-attention mechanisms, and transformer decoders. Evaluated on a synthesized dataset of 2,991 machining sequences, MP-GFormer achieves 24% and 36% accuracy improvements for main and sub-operation predictions respectively compared to state-of-the-art dynamic graph learning methods (DyGFormer and DiffPool), demonstrating that incorporating geometric domain knowledge significantly outperforms geometry-agnostic approaches in capturing the evolving dependencies between part geometries and sequential machining operations.

**Strengths:**

The paper demonstrates strong originality through its novel formulation of machining process planning as a geometry-aware dynamic graph learning problem. While dynamic graph learning has been applied to various domains, existing approaches fail to capture the essential domain constraint that part geometry fundamentally determines feasible machining operations. The key innovation lies in the creative integration of two complementary geometric representations: STL-based process graphs capturing evolving geometries after each operation, and BRep-based design graphs encoding initial part constraints. The cross-attention mechanism that fuses these representations is particularly elegant, allowing the model to learn how current geometry relates to design intent. This architecture—combining GAT for spatial relationships, cross-attention for geometric fusion, and transformer decoder for temporal dependencies—represents a thoughtful synthesis of existing techniques specifically tailored to manufacturing's unique spatio-temporal-geometric structure.
The work carries substantial practical and scientific significance. Machining process planning is a critical bottleneck in smart manufacturing, traditionally requiring extensive expert knowledge that limits automation and scalability. The reported 24-36% accuracy improvements over state-of-the-art DGL methods (DyGFormer, DiffPool) represent meaningful progress toward automated process planning systems. Beyond immediate manufacturing applications, the work establishes a broader paradigm for incorporating domain-specific geometric constraints into sequential decision-making frameworks, with potential applications to additive manufacturing, robotic assembly planning, and other domains where 3D geometry evolves through sequential operations. The approach effectively bridges machine learning research with manufacturing practice.
The paper maintains good technical quality with rigorous mathematical formulation spanning problem setup through loss functions. The experimental design includes appropriate ablation studies validating key architectural choices (GAT vs. NN encoder, decoder vs. encoder), and the t-SNE visualizations effectively demonstrate that learned embeddings achieve meaningful class separation. The paper is well-organized with clear progression from motivation through methodology to experiments, and Figure 2 effectively communicates the architecture's components and information flow.

**Weaknesses:**

The experimental validation relies entirely on synthesized data generated from only 6 parameterized base geometries (Figure 3) with 2.5-axis planar machining, severely limiting generalizability claims. Real manufacturing involves diverse multi-axis operations, material constraints, tool wear, and fixture considerations absent from this evaluation. The paper provides no validation on actual industrial process plans or analysis of synthetic-to-real transfer challenges. With only 3 main and 12 sub-operation classes and extreme class imbalance (Figure 10 shows "drilling" >3000 samples while others <100), the classification task is relatively simple. The model may be memorizing geometry-to-operation mappings for these specific parameterized variants rather than learning generalizable geometric reasoning. Testing on completely novel geometry types and real manufacturing data is essential to validate practical significance.
The benchmark comparisons are problematic because baselines (DyGFormer, DiffPool) are generic DGL methods not designed for manufacturing. More relevant domain-specific works cited in Related Works—Zhang et al. (2025) graph encoder-decoder, Wang et al. (2024) GCN-based, Hussong et al. (2025) GAT-based MaProNet, and Maqueda et al. (2025) transformer-based DeepMS—are absent from experimental comparisons. The claimed 24-36% improvements may be inflated by comparing against inappropriate baselines rather than manufacturing-aware approaches. Fair evaluation requires comparing against these domain methods on identical datasets.
Architectural design choices lack sufficient justification. Why is cross-attention optimal for fusing STL and BRep versus alternatives like concatenation, gating, or late fusion? The ablation study is incomplete, missing: (1) contribution of BRep graphs versus STL-only, (2) impact of different fusion mechanisms, (3) sensitivity to graph construction choices (edge definitions, node features), and (4) performance across varying sequence lengths and geometric complexity. Notably, the encoder variant achieves 0.98 test accuracy (Table 3) versus 0.78 for the proposed decoder, suggesting potential data leakage that receives only brief dismissal rather than thorough investigation. Joint operation prediction performs poorly (0.61-0.63) compared to separate predictions (0.78, 0.70) without analysis.
Critical implementation details are missing, hindering reproducibility: number of GAT layers, attention heads per mechanism, hidden dimensions, dropout rates, edge weighting schemes, and boundary condition handling in STL meshes. The terms "BRep" and "Design Graph" are used interchangeably without clarification. Loss curves (Figure 5) show test loss flattening or increasing after epoch 5, indicating overfitting that isn't discussed or addressed through regularization strategies.

**Questions:**

**Questions**

**Q1: Dataset Generalization**
Your 2,991 samples come from parametric variations of 6 base geometries. How many unique operation sequences exist versus geometric variations of the same sequence? Have you tested on completely novel geometry types beyond these variations? Without this, it's unclear if the model learns generalizable sequencing or memorizes geometry-operation patterns.

**Q2: Baseline Justification and Architecture**
Why exclude manufacturing-specific baselines cited in Related Works (Zhang 2025, Wang 2024, Hussong 2025, Maqueda 2025)? Provide ablation for BRep contribution—does STL-only match performance? The encoder variant achieves 0.98 vs. 0.78 for decoder (Table 3); can you show attention visualizations demonstrating this "data leakage"?

**Q3: Performance Gaps**
Joint prediction achieves 0.61-0.63 vs. 0.78/0.70 for separate predictions—why this gap? Did you address the extreme class imbalance in Figure 10, and how does per-class performance vary?

**Suggestions**

**S1: Strengthen Validation**
Include manufacturing-specific baseline comparisons, test on held-out geometry types, and provide complete implementation details (GAT layers, attention heads, dimensions). Add ablations for STL-only, fusion alternatives, and class balancing strategies.

**S2: Real-World Applicability**
Provide validation on real manufacturing data or detailed synthetic-to-real gap analysis. Characterize failure modes and performance variation with sequence length/complexity to establish practical deployment boundaries.

---

### Official Review · Reviewer_AGzU · 2025-11-01

**Soundness:** 2
**Presentation:** 3
**Contribution:** 2
**Rating:** 2
**Confidence:** 4

**Summary:**

Addressing the core issues of traditional Machining Planning (MP) relying on expert knowledge and existing Dynamic Graph Learning (DGL) methods lacking 3D geometric domain awareness, this paper proposes MP-GFormer—a dynamic graph Transformer model fused with 3D geometric information—aimed at achieving accurate prediction of machining operation sequences.
The paper first elaborates on the complexity of MP and the limitations of existing technologies, pointing out that the lack of 3D geometric information is the key reason for the insufficient performance of DGL methods in MP. It then reviews the research status of Transformer-based dynamic graph learning and the application of graph models in MP, clarifying the research gaps. Next, it details the three-stage architecture of MP-GFormer (encoder, Transformer, classifier): GAT is used to capture geometric topological features, cross-attention correlates geometry with operations, and the Transformer learns dynamic dependencies. Meanwhile, a synthetic dataset is constructed, containing STL files (post-machining geometry), BRep files (initial design), and textual operation labels. Finally, the model performance is verified through hyperparameter selection, ablation experiments, and benchmark comparisons (DyGFormer, DiffPool). Results show that MP-GFormer improves the prediction accuracy of main/sub-operations by 24% and 36% respectively compared to the best baseline, demonstrating the significant value of integrating 3D geometric information for the MP task.

**Strengths:**

Technical Innovation: Filling the Integration Gap between 3D Geometry and Dynamic Graph LearningExisting DGL methods in MP only focus on sequential graph structures, ignoring the core domain knowledge of part 3D geometry, which leads to models lacking domain awareness. MP-GFormer is the first to deeply integrate 3D geometric information into a dynamic graph Transformer framework: it captures the post-machining geometric evolution through process graph sequences converted from STL files, obtains initial geometric benchmarks from design graphs converted from BRep files, and then correlates geometric features with operation dependencies via GAT and cross-attention mechanisms. Technically, it breaks through the limitation of domain information deficiency in traditional DGL, providing a new paradigm for "geometry-operation" collaborative modeling in MP tasks.
Rationality of Method Design: Multi-module Collaboration Guarantees Prediction PerformanceThe three-stage design of the model architecture (encoder - Transformer - classifier) forms a complete "geometric input - feature processing - operation output" pipeline, with complementary functions and tight logic among modules: the encoder extracts multi-dimensional features through GAT and temporal encoding; the Transformer captures operational temporal dependencies, geometric dynamic dependencies, and their cross-dependencies via three types of attention mechanisms (masked self-attention, temporal self-attention, cross-attention); the classifier achieves accurate prediction through cross-entropy loss. Ablation experiments verify that the GAT encoder improves the main operation test accuracy by 12% compared to the NN encoder, and the Transformer decoder can avoid data leakage. These results demonstrate that each module design contributes to performance improvement, and the overall architecture has strong rationality.

**Weaknesses:**

Insufficient Model Transferability: Dependence on "Full-Operation Geometric Sequence" Training Limits Cross-Scenario ApplicationThe training of MP-GFormer in the paper relies on "the sequence of STL geometric graphs of the part after each machining operation"—that is, complete geometric evolution data of the part from raw material to finished product (geometric state corresponding to each operation) is required to complete model training. However, in practical industrial scenarios, there are significant differences in part types, machining equipment, and process preferences among different factories: for example, Factory A produces box-type parts with machining steps mostly including "milling surface - drilling - milling contour," while Factory B produces shaft-type parts with steps such as "turning - grinding - drilling," and their geometric evolution sequences are completely different. If the model trained on data from Factory A is transferred to Factory B, the lack of "full-operation geometric sequence" training data for parts in Factory B will prevent the model from adapting to the geometry-operation correlations of the new scenario, resulting in insufficient transferability and difficulty in achieving generalized application across factories and part types.
Model Complexity and Efficiency: High Computational Cost Caused by High-Dimensional Features and Multi-Head AttentionThe encoder of MP-GFormer needs to process high-dimensional features of STL graphs (8-dimensional nodes, 5-dimensional edges) and multi-attribute features of BRep graphs. The Transformer module adopts a multi-head attention mechanism and needs to handle process graphs with a sequence length T (T is the number of machining operations, which can reach dozens for complex parts), leading to high model computational complexity. The paper does not mention the time cost of model training and inference: for instance, the comparison of training time between MP-GFormer and baseline methods (e.g., DyGFormer) under the same hardware environment, and the inference time per sample. In industrial scenarios, process planning requires rapid response (e.g., real-time adjustment of machining steps on the production line). If the model's computational efficiency is too low, it may fail to meet the real-time requirements of practical applications.

**Questions:**

See Weaknesses

---

### Note · Authors · 2025-11-14

**Comment:**

We have decided to withdraw the paper due to the limited availability of real-world industrial data, which was noted as an important aspect in the reviews. We believe that addressing this limitation with additional data and validation will allow us to strengthen the study and improve its overall contribution.

**Withdrawal Confirmation:**

I have read and agree with the venue's withdrawal policy on behalf of myself and my co-authors.